# Nitrogen Loss and Migration in Rice Fields under Different Water and Fertilizer Modes

**DOI:** 10.3390/plants13050562

**Published:** 2024-02-20

**Authors:** Shurong Hao, Xia Liu, Congyi Liu, Wentan Liu

**Affiliations:** College of Agricultural Science and Engineering, Hohai University, Nanjing 211106, China; liucongyi2022@163.com (C.L.); 221610010019@hhu.edu.cn (W.L.)

**Keywords:** rice field, irrigation and drainage mods, reduced fertilization, nitrogen loss, nitrogen migration

## Abstract

Irrigating aquaculture wastewater in appropriate irrigation and drainage modes in paddy fields could reduce water and fertilizer loss. However, the precise mechanisms involved in the degradation and movement of nitrogen in various water and fertilizer modes are still not fully understood. This study involves conducting a controlled experiment using barrels to examine the effects of various water quality, irrigation and drainage methods, and fertilization levels. The aim is to analyze the patterns of nitrogen degradation, loss, migration, and absorption in surface water, underground drainage, and soil leakage at different depths. The results showed the following: (1) The paddy field has a significant purification effect on aquaculture wastewater after one day of irrigation, reached at 78.55–96.06%. (2) Aquaculture wastewater irrigation increased nitrogen concentration in the plough layer, which helps rice roots absorb nitrogen and boosts plant TN. (3) In special dry years, underground seepage is the predominant method of nitrogen loss, and underground drainage nitrogen concentration peaks 2–6 days after fertilization. (4) Under aquaculture wastewater irrigation, the TN loss load of II decreased by 27.65–42.45% than FSI. Compared with IA-80, the TN degradation rate of IA in surface water increased by 18.51%, TN loss load decreased by 5.48%, TN absorption rate significantly increased by 14.61%, and yield increased by 31.14% significantly. IA is recommended in special dry years, which can improve the TN absorption rate and ensure high yield while significantly reducing the loss load of nitrogen. The findings can provide a basis for the purification of aquaculture wastewater through paddy field ecosystems in response to fertilizer supply levels.

## 1. Introduction

Rice yield and aquaculture are essential areas in agriculture [1], playing a vital role in ensuring food security and promoting efforts to increase income levels. Presently, the dominant method employed in Chinese aquaculture is the intense mode, which is distinguished by a high breeding density and a significant feeding volume [2]. The release of aquaculture wastewater not only causes the wastage of important freshwater resources, but also leads to the emission of significant amounts of contaminants, including animal and plant remains and soluble nitrogen, into the adjacent water bodies. This exacerbates the process of eutrophication in rivers and lakes [3]. Since the 1990s, extensive exploration has been conducted on the environmental restoration technology of aquaculture wastewater at home and abroad. Among these methods, phytoremediation has gained significant attention due to its environmentally friendly nature and cost-effectiveness [4,5]. The rice cultivation region, which is a significant consumer of agricultural water, spans around 30.216 million hectares [6], and the wetland system, which utilizes rice and other crops as restorative plants, can effectively decrease the nitrogen content in aquaculture wastewater by 29.0–58.7% [7]. Therefore, employing rice fields to absorb aquaculture wastewater can not only facilitate the recycling of nutrients [8], but also substantially mitigate the threat of agricultural non-point source pollution [9], which holds crucial implications for the establishment of ecological irrigation zones.

The flat region in southern China possesses exceptional soil and water resources, making it very conducive to rice agriculture and aquaculture. Nevertheless, it has encountered numerous challenges in recent times. On the one hand, the southern rice-producing area experiences an irregular distribution of rainfall, rendering it susceptible to both seasonal droughts and floods. Rice cultivation consumes about 65% of the total agricultural water consumption [6], with a water resource utilization rate of approximately 40%, which is much lower than the 70% recorded in industrialized nations [10]. Implementing appropriate irrigation and drainage strategies is crucial for minimizing irrigation water usage. The practice of rice planting suggests the use of water-saving irrigation modes, including control irrigation [11,12], water-catching and controllable irrigation [13], and intermittent irrigation [14,15]. Rice growth and development depend on the presence of nitrogen, which is a vital nutrient [16]. At present, the typical quantity of nitrogen fertilizer applied in rice fields is approximately 180 kg/hm^2^ [17], but the efficiency of nitrogen fertilizer absorption and utilization in rice fields is below 30% [18]. Excessive fertilization results in nitrogen loss, which in turn leads to difficulties such as increased algal reproduction, changes in species composition, degradation of water biological structure, and other related issues [19]. It has been shown that different irrigation and drainage modes can effectively improve the water utilization rate and reduce the loss of nitrogen and phosphorus by reducing the lower limit of irrigation [12], and increasing the water storage depth after rain [13] in the field. Williams [20] showed that control irrigation could reduce concentration of NO_3_^−^-N about 20%, compared with traditional irrigation mode. Guo [13] discovered that the TN loss decreased by 35% than frequent and shallow irrigation.

On the other hand, aquatic animals only absorb one-third of the nitrogen present in their feed [3,21,22]. The remaining unabsorbed nitrogen is retained in aquaculture wastewater and sediment [23], which is subsequently discharged into the river to cause environmental pollution together with the aquaculture wastewater. Li [24] found that it can nevertheless fulfill the nutrient requirements of rice when used in aquaculture wastewater, under 80% of nitrogen conventional fertilizer. Meanwhile, Zhou [22] and Yan [3] discovered that 80% of the standard fertilization treatment resulted in comparable yield levels to traditional fertilization, and the yield was still above 9000 kg/hm^2^, without the potential risk of reduced yield. Therefore, the irrigation of paddy fields with aquaculture wastewater could satisfy the demand for rice of water and nutrients, reduce the use of water and fertilizing amount, which means it has important economic benefits. Additionally, it achieves removal rates of 16.1–32.4% for TN, 2.9–33.4% for NH_4_^+^-N, and 12.6–92.5% for NO_3_^−^-N [25]. This indicates that the use of aquaculture wastewater to irrigate paddy fields has good environmental benefits.

Many researchers, both domestic and international, have conducted numerous studies on paddy fields that are irrigated with aquaculture wastewater; however, there are relatively few studies on the mechanism of degradation, migration, and transformation effect of nitrogen in different water and fertilization modes. This study examines nitrogen concentration in surface water, underground drainage, and soil leakage at different depths, as well as its impact on rice yield, by considering water quality, fertilizing amount, and irrigation and drainage modes as the main determinants. We conduct an analysis of the processes of nitrogen degradation, loss, migration, and absorption in paddy fields. The objective is to promote the recycling of water resources and minimize nitrogen loss, in order to establish a scientific foundation and reference for the widespread use and application of aquaculture wastewater.

## 2. Materials and Methods

### 2.1. Description of the Trial Area

The experiment was conducted from May to October 2022 in Water-saving Park (31° 54′57″ N, 118°46′37″ E) of Jiangning Campus, Hohai University, Nanjing, Jiangsu Province. The area has a subtropical humid climate with average annual rainfall of 1106.5 mm, average annual evaporation of 900 mm, and average annual temperature of 15.5 °C. The test soil was obtained from the clay loam with the dry bulk weight of 1.31 g/cm^3^, saturated water content of 38.7%, pH of 6.68, organic matter, total nitrogen, total phosphorus, effective phosphorus, and hydrolysis nitrogen are 21.17 g/kg, 1.06 g/kg, 0.52 g/kg, 22.34 mg/kg, and 105.27 mg/kg, respectively.

Figure 1 illustrates the variations in daily precipitation and temperature in different periods of rice growth. The cumulative rainfall of different periods (27 June to 23 October) was only 137.3 mm, which was a special dry year. The number of rainfall days were 20 days, and the longest accumulated 28 days with no rainfall was from 31 July to 25 August. The maximum daily rainfall was 31.6 mm (10 October), followed by 26.2 mm (30 July), and other daily rainfall was less than 18.5 mm. The rainfall was mainly distributed as 38.9% in ripening stage and 36.7% in tillering stage.

### 2.2. Trial Design

The barrel measured 40 cm by 40 cm by 100 cm in length, width, and height, according to the barrel test. The underground drainage level observation pipe and 3 soil leakage water holes were set on the outer wall of the barrel. The water holes were 18 cm, 36 cm, and 54 cm away from the soil table, respectively. After being dried, broken, and screened, the soil samples were compacted and packed into the barrel. The bottom of the barrel was filled with a 20 cm thick sand and gravel cushion, the top of barrel was set aside for water storage at a depth of 20 cm, and the underground drainage extraction valve of the cushion was filled with 60 cm deep soil. The arrangement of the site and the structure of experiment barrel is shown in Appendix A.

For the trial set “3 factors level 2”, with a total of 6 treatment, each treatment repeated thrice, we chose frequent and shallow irrigation (FSI) and intermittent irrigation (II) as the irrigation and drainage mode; clear water (CW) and aquaculture wastewater (AW) as irrigation water; normal fertilization (NF) and 80% of the conventional fertilization (NF-80). The experimental design is depicted in Table 1, and different irrigation and drainage mode moisture control standards are described in Appendix A.

The trialed rice is “Nanjing 9108”, which was raised on May 18, transplanted on June 27, and harvested on October 23. We transplanted three-leaf seedlings of rice, with 15 cm by 15 cm, 4 holes per barrel, and 3 plants in each hole. The fertilizers urea (including 46% N), perphosphate (including 15% P_2_O_5_), potassium chloride (including 63% K_2_O) were used in this test 3 times, applying base fertilizer on June 26, tillering fertilizer on July 7, and ear fertilizer on August 23. The segmentation of the birth period and the quantity of fertilization are shown in Appendix A. After transplanting, different treatments irrigated clear water or aquaculture wastewater, respectively. The aquaculture wastewater adopted the fish feed fermentation method, for which the concentration ranges of TN, NH_4_^+^-N, NO_3_^−^-N, and TP were 5.0–6.0 mg/L, 3.0–4.0 mg/L, 0.7–0.9 mg/L, and 0.9–1.2 mg/L, respectively. During the whole growth period, all treatments were the same except for irrigation and drainage measures, irrigation water, and fertilization amount.

### 2.3. Determination Index and Methods

In this test, the content of total nitrogen (TN), nitrate nitrogen (NH_4_^+^-N), and ammonium nitrogen (NO_3_^−^-N) in surface water, underground drainage, and soil leakage were observed. The underground drainage was collected every 2 d during the regreening period, every 3–4 d from the tillering period, and measured before and after each fertilization. Surface water was collected after flowering period (4 September) for 8 h, 1 d, and 2 d. Soil leakage from upper soil (−18 cm), middle soil (−36 cm), and lower layer (−54 cm) was measured during each growth period. TN, NH_4_^+^-N, and NO_3_^−^-N concentrations were determined by alkaline potassium persulfate digestion UV spectrophotometry, Nessler’s reagent colorimetric method, and phenol disulfonic acid spectrophotometry, respectively. After rice harvest, the TN content of plants was determined by determination of total nitrogen in the plant by the Kjeldahl method. During the birth period, the water depth of the field in the barrel was measured every day, and the rainfall depth was recorded. The irrigation and drainage were conducted according to the upper limit and lower limit of irrigation and rain storage during different fertility periods, and the irrigation amount and surface and underground drainage during the whole birth period were recorded.

The calculation of nitrogen reduction rate(θ_1_), TN loss rate, TN content, and TN absorption rate are as follows:$$\begin{array}{c} \theta_{1}=\left(C_{t}{\;-\;C}_{t-1}\right){/C}_{0}\;\times \;100\\ \theta_{2}{=W}_{l}{/W}_{f}\;\times \;1000\\ \omega =\left(m_{k}\beta_{k}{\;+\;m}_{l}\beta_{l}{\;+\;m}_{h}\beta_{h}\right){\;\times \;10}^{-6}\\ {\gamma =\omega /W}_{f}\times 100 \end{array}$$

θ_1_ is the degradation rate of nitrogen in the paddy field, %; C_t_ is the concentration of nitrogen in surface water at t time after irrigation and taken for 8 h, 1 d, 2 d, mg/L; C_t−1_ is the nitrogen concentration in surface water at the last mount of t time after irrigation and taken for 0 h, 8 h, 1 d, mg/L; C_0_ is the concentration of nitrogen in surface water 0 h after irrigation, mg/L; θ_2_ is the TN loss rate, ‰; W_l_ is the TN loss load, kg/hm2; W_f_ is the TN application, kg/hm^2^; ω is the TN content of the plants, kg/hm^2^; m_k_ is the kernel dry mass, kg/hm^2^; β_k_ is the TN content in kernel, mg/kg; m_l_ is the leaves dry mass, kg/hm^2^; β_l_ is the TN content of leaves, mg/kg; m_h_ is the haulm dry mass, kg/hm^2^; β_h_ is the TN content of haulm, mg/kg; γ is the absorption rate of TN, %.

### 2.4. Data Analysis

Data analysis was performed by Microsoft Excel 2023, graphed by GraphPad Prism 9.5.0, and one-way analysis of variance (ANOVA) was performed for the general linear model of SPSS 13.0 software (SPSS, Chicago, IL, USA). Significance was tested using Duncan (*p* < 0.05) for multiple comparisons.

## 3. Results

### 3.1. Degradation Rate of the Surface Water after Irrigation

Figure 2 displays the nitrogen concentration in surface water over a 2 day period following irrigation in various treatments. Table 2 presents the decomposition rate of nitrogen. The TN, NH_4_^+^-N, and NO_3_^−^-N concentrations reduced to a range of 0.189–0.224 mg/L, 0.120–0.166 mg/L, and 0.059–0.198 mg/L, respectively. The degradation rates for TN, NH_4_^+^-N, and NO_3_^−^-N were determined to be between 96.06–96.67%, 95.28–96.59%, and 74.71–92.46%, respectively. The findings suggest that the paddy field has a significant capacity for purifying aquaculture wastewater, with substantial degradation of TN and NH_4_^+^-N, followed by NO_3_^−^-N.

The TN degradation rate of AW in rice fields demonstrated an initial phase of rapid degradation, succeeded by a following phase of slower degradation. The TN degradation rate of AW was 82.33–87.19% within the first 0–8 h. After 8 h, the degradation rate of AW fell significantly to only 9.49–13.72%, indicating that the TN degradation effect was concentrated within 8 h after irrigation, and this period is a high-risk period of TN loss in surface water. Within 8 h of the high-risk period of TN loss, FA and FA-80 were significantly higher than IA and IA-80, but the difference between each irrigation mode within 2 d after irrigation was not significant. The TN degradation rate of AW from 0 h–2 d in paddy fields (ranging from 96.06–96.67%) was significantly higher than CW (ranging from 44.42–48.39%), while the difference between treatments (FA, FA-80, IA, IA-80) was not significant. The NH_4_^+^-N degradation rule of AW was similar to the TN. The TN and NH_4_^+^-N degradation rate of CW changed little with time, and the degradation rate was significantly lower than FSI.

The NO_3_^−^-N degradation rate of AW was significantly greater than CW. The NO_3_^−^-N degradation rate of AW was significantly slower after 1 d, and the degradation rate of 0 h–1 d was 78.55–94.13%, indicating that the degradation effect of NO_3_^−^-N was mainly concentrated within 1 d after irrigation, and this period is a high-risk period of NO_3_^−^-N loss in surface water. During the high-risk period, the NO_3_^−^-N degradation rate of IA-80 was significantly lower than FA, FA-80, and IA. However, the NO_3_^−^-N degradation effect of CW is reflected after 8 h. The NO_3_^−^-N concentration of CW increased from 0 h to 8 h, and decreased within 8 h to 2 d (degradation rate ranging from 29.92–45.3%). At 2 d after irrigation, the NO_3_^−^-N concentration of CW (ranging from 0.416–0.487 mg/L) is significantly higher than FW (ranging from 0.059–0.198 mg/L).

### 3.2. Nitrogen Concentration in Underground Drainage

Figure 3 illustrates the variation in nitrogen concentration in the subterranean drainage for each treatment. The nitrogen concentration in underground drainage was mainly affected by fertilization, and the peaks after three times of fertilization showed a downward trend, indicating that the influence of base fertilizer and tillering fertilizer on the nitrogen concentration in underground drainage was greater than heading fertilizer. The content of NH_4_^+^-N accounted for more than 50% of TN, indicating that the nitrogen loss in underground drainage is mainly NH_4_^+^-N.

The highest value of TN concentration in each treatment occurred on the fourth day after the base fertilizer, and the TN concentration of FA (6.72 mg/L) was significantly higher than other treatments (ranging from 5.50–5.75 mg/L). On the fourth day after tillering fertilizer, the TN concentration of FSI reached the second peak, and the peak of intermittent irrigation was delayed by two days. The third peak was reached on the second day after the spike fertilizer. After the third peak, TN concentration fluctuated continuously, which may be caused by regular irrigation. The change pattern of NH_4_^+^-N concentration is consistent with TN. The second peak of NH_4_^+^-N concentration was on the fourth day after the tiller fertilizer. The NH_4_^+^-N concentration of II remained high during the tillering stage.

NO_3_^−^-N decreases first and rises rapidly, and reached a peak on the fifth day. The second peak was reached after tillering fertilizer, where the NO_3_^−^-N concentration decreased rapidly in each treatment, and the NO_3_^−^-N concentration of II fluctuated until the end of the tillering stage. Compared with the concentrations of TN and NH_4_^+^-N, the third peak of NO_3_^−^-N was before fertilizer application (August 16), and the second peak of TN, NH_4_^+^-N, and NO_3_^−^-N concentrations under FSI was also two days earlier than II.

The concentration of TN and NH_4_^+^-N in FA-80 was the lowest in the whole birth period between three treatments of FSI. In addition, the concentration of nitrogen in IA was the lowest in the whole birth period between three treatments of II. Although part of the nitrogen concentration in the underground drainage comes from the freshly irrigated field water, most of it still comes from the release of nitrogen in the soil profile, so this may be related to the amount of fertilization in the early stage.

### 3.3. Nitrogen Loss load Analysis

The water quantity and underground displacement were significantly lower than that of shallow irrigation. In the year 2022, there was a significant drought, with the surface water level for the entire year remaining below the maximum capacity for rain storage, and the surface displacement was 0. The nitrogen in the paddy field was mainly lost by underground drainage. The irrigation and drainage volume of different treatments is shown in Appendix A.

Figure 4 displays the nitrogen loss load of different treatments. The nitrogen loss load of FSI was considerably greater than II, with TN, NH_4_^+^-N, and NO_3_^−^-N loss loads being 1.27–1.81, 1.23–1.90, and 1.04–1.50 times higher than II, respectively. AW does not increase the risk of nitrogen loss. Under AW, the nitrogen loss load of II decreased by 33.30–47.20%, and the loss rate decreased by 27.64–42.52% than FSI. There was no significant difference between nitrogen loss load of FW and FA; nitrogen loss load under IW was significantly higher than IA, and AW had little impact on FSI, but could nitrogen loss load of IA-80. The nitrogen loss load of FA was significantly higher than FA-80, and the nitrogen loss load of IA was significantly lower than IA-80, indicating that FA-80 can reduce nitrogen flow, while IA would not increase the risk of nitrogen loss. The tillering stage is the key period of nitrogen loss, and the nitrogen loss accounted for 24.12–41.37% in the total growth period.

### 3.4. Nitrogen Distribution of Soil Leakage at Different Depths

Figure 5 illustrates the alteration in the mean nitrogen concentration across the soil vertical profile. The TN average concentration of AW was gradually decreasing, and the TN concentration at −18 cm (ranging from 0.817–0.828 mg/L) was slightly lower than NF-80 (ranging from 0.765–0.786 mg/L), but the difference was not obvious at −54 cm. The average concentration variation pattern of NH_4_^+^-N in the soil vertical profile is similar to the TN. Under AW, there was no significant difference between irrigation and drainage patterns at −18 cm, but the NH_4_^+^-N concentration of II at −54 cm (ranging from 0.322–0.330 mg/L) was less than FSI (ranging from 0.346–0.353 mg/L). The difference between the NH_4_^+^-N concentration of NF-80 and normal fertilization with soil depth was not obvious. The NO_3_^−^-N average concentration of AW decreased in the vertical profile of soil, while the average concentration of NO_3_^−^-N in CW irrigation showed little change. The NO_3_^−^-N concentration of II was less than FSI, and the NO_3_^−^-N concentration of FSI at −54 cm (ranging from 0.151–0.164 mg/L) was significantly lower than II (ranging from 0.199–0.218 mg/L). The difference between the NO_3_^−^-N concentration of NF and NF-80 with soil depth was not obvious. There was a significant difference between CW and AW. The nitrogen concentration of AW was significantly higher at −18 cm, and significantly lower at −54 cm than CW. In the lower soil, NH_4_^+^-N concentration was greater than NO_3_^−^-N, the loss with NH_4_^+^-N as the main form. The mean DO concentration change in the soil vertical profile is shown in Appendix A. The DO concentration decreased gradually in the soil vertical profile. The average DO concentration of II at −18 cm (7.17–7.28 mg/L) was slightly greater than FSI (6.64–7.28 mg/L). The DO concentration of II at −54 cm (5.54–5.87 mg/L) was significantly greater than FSI (4.46–4.77 mg/L).

### 3.5. Loss and Absorption Rate of Nitrogen

Table 3 displays the TN loss rate and absorption rate for various treatments. The fertilizing amount of FA-80, IA-80, FW, and IW is comparable, indicating that aquaculture wastewater irrigation could achieve 20% TN fertilizing amount. AW could significantly reduce the loss rate of TN and increase the TN content of the plant by 3.06–12.69%, approximately. There was no significant difference between FA and FW, but the loss rate was significantly lower than FW, and the TN content and absorption rate were significantly higher than FW. The drain loss load of IA and the loss rate were significantly lower than IW, although the nitrogen absorption rate was not significantly different, and the TN content in plants was significantly higher than IW. Different irrigation and drainage modes have different results under NF-80. There was no significant difference in TN absorption rate of FA-80 and FA, but TN loss load of FA-80 significantly reduced by 11.85% than FA. In addition, the TN loss load and loss rate of IA significantly reduced by 5.33% and 22.58%, respectively, compared with IA-80, and the TN content and nitrogen absorption rate in plants significantly increased by 28.72% and 12.75%, respectively.

The yield is from large to small in the following order: FA > FW > FA-80 > IW > IA > IA-80. The yield of FSI was significantly greater than II, and the yield of NF was significantly greater than NF-80. The yield of FA was the highest, being 13.50% higher than FA, but TN loss load was also the largest. Compared with FA, yield of FA-80 had decreased, but TN loss load decreased. Compared with IA-80, yield of IA had increased, and TN loss was significantly decreased.

## 4. Discussion

### 4.1. Nitrogen Degradation

Surface drainage is the main route of nitrogen loss in rice fields [26]. The results of this study showed that the 1 d degradation rate of TN, NH_4_^+^-N, and NO_3_^−^-N in aquaculture wastewater after irrigation reached, respectively, 88.22–96.06%, 88.92–93.55%, and 78.55–94.13%, indicating that rice field has a strong degradation effect on aquaculture wastewater, which is consistent with the conclusion of Chen [27]. Meanwhile, the nitrogen degradation rate of AW in paddy fields was significantly greater than CW, and the degradation rate of TN and NH_4_^+^-N were 3.51–4.06 and 3.77–4.07 times within 1 d; this is because the activity of microorganisms is better in the aquaculture wastewater of high nitrogen concentration, which has a better reduction effect on nitrogen [28,29]. On the other hand, denitrification is the main way of nitrogen reduction [4], and aquaculture wastewater contains a large amount of organic carbon denitrification to provide a sufficient carbon source and enhances microbial activity and metabolic function [8]. Therefore, lots of NO_3_^−^-N, transformed by NH_4_^+^-N, increased the trend of denitrification [30], which promoted the denitrification. The test results showed that the degradation rate of TN and NH_4_^+^-N had the best effect within 8 h after irrigation, while the best effect of NO_3_^−^-N was within 8 h to 1 d after irrigation (Figure 2); this is because aquaculture wastewater carries a large number of microorganisms affecting the dissolved oxygen amount in surface water, which changes the rate of nitrification and denitrification reaction [31]. Within 8 h, the degradation rate of TN reached 82.33–87.19%, and the degradation rate of NH_4_^+^-N was 80.39–91.36%. The irrigation disturbance resulted in an increase in the dissolved oxygen content of surface water due to the significant influx of air. The addition of topsoil increased microbial activity [32], and aquaculture wastewater was carried into the nitrification bacteria [33]; the coordinated action of the three promoted the nitrification reaction within 8 h. Under CW, with high oxygen content and strong nitrification, the concentration of NO_3_^−^-N did not decrease within 8 h. While the aquaculture wastewater has low oxygen content and strong denitrification, the NO_3_^−^-N concentration is constantly decreasing, which is just the opposite of CW. Denitrification was mainly concentrated during 8 h–1 d in surface water, and this is because sufficient oxygen in the first 8 h inhibited denitrification. NO_3_^−^-N, as the main product of nitration, gradually increased within 8 h after irrigation (Figure 2c). With the decrease in dissolved oxygen content in surface water and the enrichment of NO_3_^−^-N from 8 h to 1 d after irrigation, the denitrification gradually increased [21], and the degradation rate of NO_3_^−^-N reached 78.55–94.13% within 1 d after irrigation; there is a high-risk period of nitrogen runoff loss within 1 d after irrigation, when draining should be avoided to reduce the risk of pollution in agricultural fields.

FSI had a higher lower limit of irrigation than II. As a result, the field experienced a flooded state with lower water and soil oxygen concentrations, leading to the activation of reductase activity [8] and enhanced denitrification. Consequently, TN degradation rate was higher. Within 8 h after irrigation under NH_4_^+^-N, the degradation rate of NF-80 was significantly higher than NF, mostly because of the low nitrogen absorption capacity of rice within this short timeframe. The saturation of NH_4_^+^-N in aquaculture wastewater led to limited absorption rates of NH_4_^+^-N concentrations during nitration within a specific range. Therefore, reducing the amount of fertilizer can enhance the degradation and absorption of NH_4_^+^-N, aligning with the findings of Wang [34]. There was no significant difference in the TN and NH_4_^+^-N degradation rate of 2 d between different irrigation and drainage modes and fertilization amounts, which may be related to the precipitation of soil, filtration, and degradation of microorganisms [34], indicating that the hydraulic retention time of aquaculture wastewater in rice fields is more critical to nitrogen degradation.

### 4.2. Nitrogen Migration

The nitrogen migration at different depths has the ability to impact both the fertility of the soil and the quality of groundwater. This study discovered that quality of irrigated water had a notable impact on the distribution of nitrogen in the soil at various depths. The results indicated that the nitrogen concentration under AW decreased significantly as the soil depth increased. This suggests that paddy fields can effectively capture and break down nitrogen in aquaculture wastewater. Furthermore, the nitrogen is primarily concentrated in the tillage layer, which is beneficial for the absorption and utilization of rice roots. These findings align with the findings of Yan [3]. The NH_4_^+^-N concentration of AW decreased significantly with the increase in soil depth, because the positive electricity of NH_4_^+^-N is easy to be intercepted by the negative electric soil. Furthermore, a large number of microorganisms in aquaculture wastewater formed biofilm in the surface rhizosphere [21] and effectively intercepted NH_4_^+^-N in the upper soil layer. CW lacks the interception of biofilms constructed by microorganisms, so it is less effective against NH_4_^+^-N than AW. The NO_3_^−^-N concentration of AW decreased significantly with the increase in soil depth, which is related to the weakening of nitrification with the increase in soil depth.

The nitrogen loss load of II was significantly lower than FSI (Figure 4); nitrogen was lost through underground leakage in special dry year, indicating that TN, NH_4_^+^-N and NO_3_^−^-N of II migrated less to the lower soil layer than FSI. This is because FSI has a larger storage depth. Under the action of water pressure, it would promote the downward migration of more nitrogen [35]. Although TN and NO_3_^−^-N concentrations of II were greater than FSI at different depths (Figure 5), the nitrogen loss of II was still lower than FSI (Figure 4); this may be related to the smaller leakage of II than FSI. The NH_4_^+^-N concentration of II was significantly less than FSI in the lower layers; this is because FSI maintained a prolonged presence of water in the field, and large leakage water occupied the soil pores in the lower layers, resulting in a notable decrease in oxygen levels in the middle and lower layers of the soil compared to II. Nitrification and denitrification in deep soil are mainly affected by oxygen concentration [36]. Kim [37], Fujii [38], and others discovered that nitrification was nearly halted as a result of very low DO concentration in non-rhizosphere soil, so the NH_4_^+^-N concentration of FSI in lower soil was significantly higher than II. The nitrogen concentration of NF and NF-80 under AW was only significantly different in the upper layer. This suggests that the amount of fertilization under AW significantly affected the soil nitrogen concentration in the upper layer, which was consistent with the results of Wiatrak [39].

### 4.3. Nitrogen Loss

Underground leakage is a primary pathway for nitrogen loss [40]. This study founded that surface runoff is reduced in dry years (Appendix A), and nitrogen is frequently lost by the form of underground seepage. The loss of underground nitrogen leakage in rice fields is affected by both nitrogen concentration and leakage amount. The test findings indicated that nitrogen loss during the tillering stage accounted for 24.12 to 41.37% of the whole growth period. On the one hand, 70% of the nitrogen fertilizer in the whole growth period was applied with the base and tillering fertilizer; thus, the nitrogen concentration of underground drainage during the tillering period was relatively high. On the other hand, 36.7% of the rainfall was concentrated in the tillering period, resulting in a large underground leakage in this period, resulting in the tillering period becoming the critical period of nitrogen loss, indicating that the tillering stage is the main period of nitrogen loss, which is consistent with the findings of Zheng [41].The nitrogen concentration of underground drainage peaked at 2 to 6 d after fertilization (Figure 3), indicating that it could be a high risk period of nitrogen loss after fertilization, and it should be avoided to fertilize within a week before rain, similar to the conclusion of Lv [42]. There is no significant difference in nitrogen concentration between each treatment after the jointing-booting period (Figure 3), indicating that it is not responsible for nitrogen concentration of underground drainage under different water and fertilization modes, which is consistent with the findings of Yan [3]. Although part of the nitrogen concentration in the underground drainage comes from the freshly irrigated surface water, most of it still comes from the release of nitrogen in the soil profile, so this may be related to the amount of fertilization in the early stage.

The total loss load of TN, NH_4_^+^-N, and NO_3_^−^-N in II was significantly lower than FSI, because reduction in leakage is a key factor for reduced II nitrogen loss, indicating that the irrigation mode of alternating dry and wet could reduce nitrogen loss, which is consistent with the results of Katsura [43]. The results of this test showed that AW does not necessarily have a negative impact on nitrogen loss, and the total loss load of nitrogen in AW is reduced by 4.14–47.77% compared with CW. This is because the nitrogen brought by aquaculture wastewater is not concentrated at one time, but carried in a small amount by irrigation many times, and the nitrogen is mainly enriched in the 0 to −18 cm tillage layer, which is easy to be absorbed and utilized by rice. It may be that a large number of microorganisms and biomass in aquaculture wastewater form a biofilm in the surface soil [21], which produces a venue for the strong nitrification to transform NH_4_^+^-N into NO_3_^−^-N absorbed easily in rice, promoting nitrogen degradation and absorption by plants [44].

This study showed that there was no significant difference in underground drainage between normal and reduced fertilization (Appendix A), which means different water and fertilization modes are not responsible for nitrogen concentration in underground drainage, which is consistent with results reported by Lian [45]. FA-80 nitrogen concentration in underground drainage was less than FA, and TN loss load decreased by 19.24%; IA nitrogen concentration in underground drainage was less than IA-80, and TN loss load decreased by 5.48%.

### 4.4. Nitrogen Absorption

Different fertilization and irrigation drainage modes alter the nitrogen absorption in rice [46]. Nitrogen plays a crucial role in various metabolic processes, such as the incorporation of ammonia and the production of amino acids in crops [43]. Proper nitrogen application can facilitate nitrogen transport in nutritional organs and improve the efficiency of nitrogen absorption and utilization [43]. The TN content of FA-80 in plants was significantly lower than FA, but the difference in TN absorption rate between the two treatments was not significant, which indicated that TN content had a positive correlation with the fertilizing amount under FSI. However, the TN absorption rate did not increase with the increasing fertilizing amount of nitrogen (Table 3), when the amount of nitrogen was beyond certain limits, which is consistent with the conclusion of Zhang [47]. IA had higher nitrogen content and nitrogen absorption rate than IA-80; this is because light drought promoted root growth, which improved the nitrogen absorption efficiency [48]. The restoration of water after a drought led to a large enhancement in the activity of nitrogen metabolism enzymes. This, in turn, boosted nitrogen accumulation and operation, and resulted in a significant improvement in the nitrogen utilization rate. The nitrogen absorption rate was also maintained at a high level when the nitrogen fertilizing amount of II exceeded certain limits. Our research suggests that the alternating application of dry and wet nitrogen fertilizer, in addition to regular nitrogen fertilizer, enhances the uptake of nitrogen and improves the efficiency of nitrogen fertilizer utilization in rice. These results are in line with those reported by Xu [49], who showed that proper reduced fertilization and irrigation with aquaculture wastewater would not affect the transfer of nitrogen to plants. The present study showed that, nitrogen is mainly enriched in the tillage layer under AW, which is easy to be absorbed and utilized by rice, significantly improving the nitrogen absorption rate and nitrogen content of rice, while the rate of nitrogen leaching is significantly reduced. The TN absorption rate of IA increased by 14.61%, which was significantly more than IA-80. Meanwhile, yield increased significantly by 31.14%, and nitrogen loss load decreased by 5.48%. Considered comprehensively, IA is recommended in special dry years. By optimizing the nitrogen absorption rate of aquaculture wastewater in paddy fields, the yield of rice can be increased while reducing nitrogen loss. This improvement allows for a large reduction in the volume of clear water needed for irrigation, saving around 11,586.9 m^3^/hm^2^.

### 4.5. Prospects and Shortcoming

Aquaculture wastewater irrigation for farmland soil input a large number of organic matter, which enhance microbial activity and further strengthen nitrogen degradation. Microorganisms are an important factor driving the soil nitrogen cycle process. Under different water modes, microbial community diversity, structure, and functional activity of rice have influence on nitrogen degradation, absorption and transformation process. The barrel test in this study was exploratory. Although there is no detailed exploration in microorganisms, the conclusions obtained in this paper can provide a reference for subsequent studies. The best water and fertilizer model recommended in this study is the conclusion obtained under the special dry year. In that year, the low soil moisture content increases the soil aeration, and the large irrigation amount of aquaculture wastewater brings more nitrogen, which makes the soil water and fertilizer conditions different from the other hydrology year type. Therefore, multi-hydrological annual tests should be carried out, and the effects of soil redox potential and microbial activity on nitrogen loss in rice should be taken into account. The response mechanism of rice growth, nitrogen degradation, and nitrogen loss in different water and fertilizer modes should be clarified, so as to provide sufficient theoretical basis for recommending the best water and fertilizer mode.

## 5. Conclusions

This study investigated the effects of different water and fertilizer modes on nitrogen degradation, loss, migration, and absorption rules of paddy fields. The main conclusions are as follows:(1)The paddy field has a significant purification effect on aquaculture wastewater. After one day of irrigation, the nitrogen degradation rate reached 78.55–96.06%, illustrating that there is a high-risk period of nitrogen runoff loss during 1 d, when surface drainage should be avoided to reduce the risk of pollution in agricultural fields.(2)The water and fertilizer mode affects nitrogen migration. Nitrogen with irrigation on aquaculture wastewater is mainly enriched in the plough layer, which is more conducive to the absorption and utilization of rice roots. FSI is more likely to produce nitrogen migration to the lower soil than II.(3)In special dry years, underground seepage is the main mode of nitrogen loss. The peak of nitrogen concentration in underground drainage of rice fields occurred 2 to 6 d after fertilization, so fertilization should be avoided within one week before rain.(4)Irrigation on aquaculture wastewater brought more nitrogen into the rice field, but did not increase nitrogen loss and significantly increased the TN content of plants. Under aquaculture wastewater irrigation, the TN loss load of II decreased by 27.65–42.45% compared to FSI. Compared with IA-80, the TN degradation rate of IA in surface water increased by 18.51%, TN loss load decreased by 5.48%, TN absorption rate significantly increased by 14.61%, and yield increased by 31.14% significantly.

Considering the rules of nitrogen degradation, loss, migration, and absorption, the mode of irrigation on aquaculture wastewater with intermittent irrigation and normal fertilization is recommended in special dry years, which can improve the utilization rate of nitrogen absorption in rice, ensure high yield, and significantly reduce the nitrogen loss.

## Figures and Tables

**Figure 1 plants-13-00562-f001:**
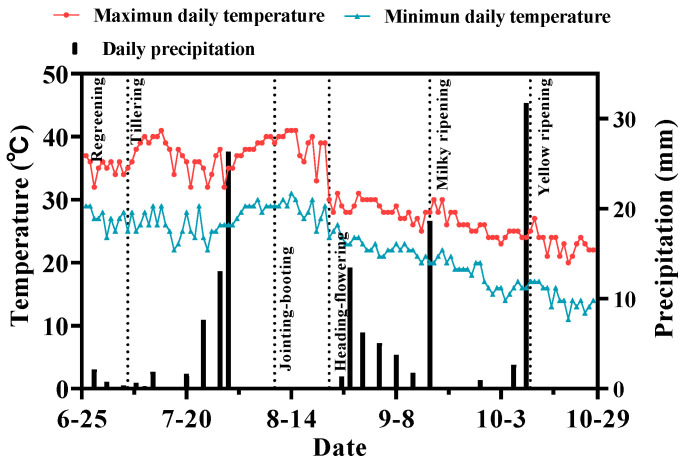
The daily precipitation and temperature change in the different periods of rice growth.

**Figure 2 plants-13-00562-f002:**
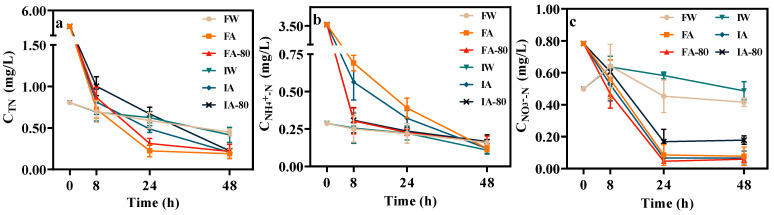
Nitrogen concentration of surface water during two days on September 4 in different treatments after irrigation (**a**–**c**) show the concentration of TN, NH_4_^+^-N, and NO_3_^−^-N in each treatment field over time.

**Figure 3 plants-13-00562-f003:**
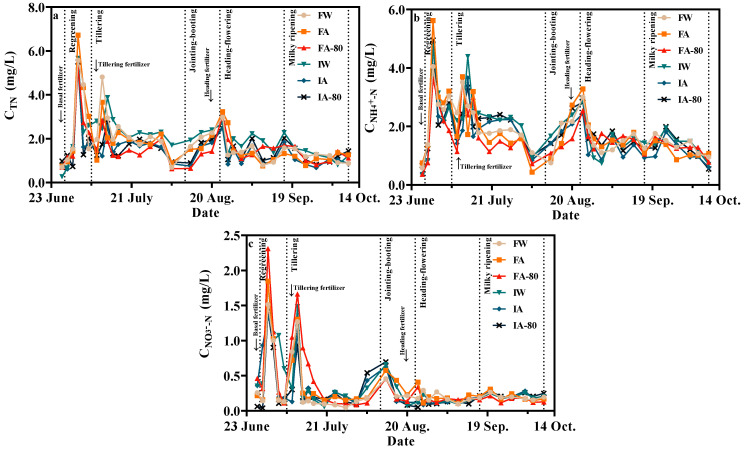
Change of nitrogen concentration in underground drainage of different treatment: (**a**–**c**) indicate the concentration changes of TN, NH_4_^+^-N, and NO_3_^−^-N, respectively.

**Figure 4 plants-13-00562-f004:**
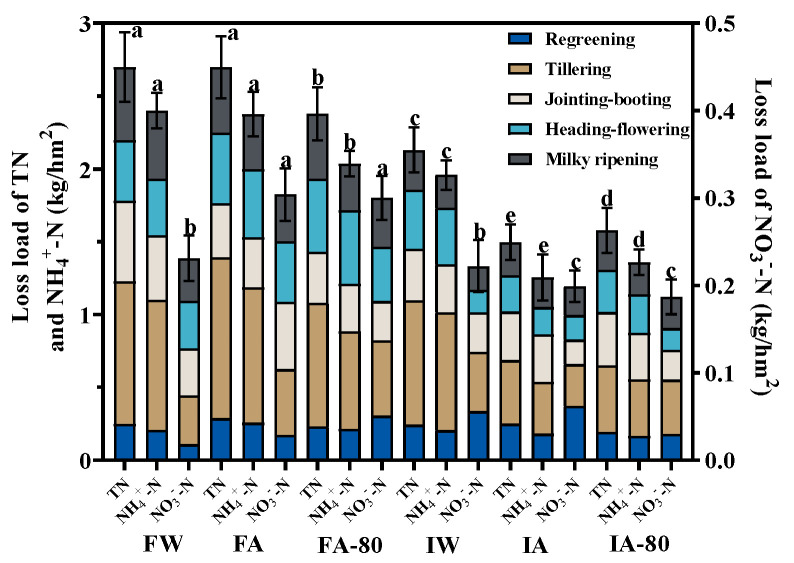
Nitrogen loss load in different treatments: the letters in the graph indicate the significance of differences between different treatments of the same nitrogen form (*p* < 0.05).

**Figure 5 plants-13-00562-f005:**
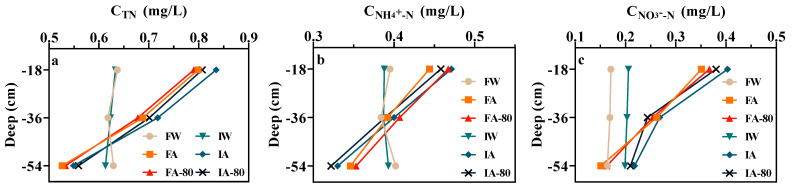
Change of nitrogen concentration in soil vertical profile: (**a**–**c**) indicate the concentration changes of TN, NH_4_^+^-N, and NO_3_^−^-N, respectively.

**Table 1 plants-13-00562-t001:** Experimental design.

Irrigation and Drainage Mode	Clear Water (CW)	Aquaculture Wastewater (AW)	
	Normal Fertilization (NF)	Normal Fertilization (NF)	80% of the Conventional Fertilization (NF-80)
FSI	FW (FSI + CW + NF)	FA (FSI + AW + NF)	FA-80 (FSI + AW + FN-80)
II	IW (II + CW + NF)	IA (II + AW + NF)	IA-80 (II + AW + FN-80)

**Table 2 plants-13-00562-t002:** Degradation rate in surface water after 2 days of irrigation in different treatments.

Nitrogen Form	Time after Irrigation	Degradation Rate of Nitrogen (%)					
		FW	FA	FA-80	IW	IA	IA-80
TN	0 h–8 h	13.52 ± 1.23 d ^1^	87.19 ± 3.06 a	84.71 ± 4.08 b	14.27 ± 1.92 d	85.68 ± 3.68 b	82.33 ± 3.33 c
	8 h–1 d	13.40 ± 1.05 a	8.87 ± 0.98 bc	9.78 ± 1.12 b	8.19 ± 1.26 c	5.74 ± 0.56 d	5.89 ± 0.58 d
	1 d–2 d	17.49 ± 1.44 b	0.62 ± 0.12 f	1.67 ± 0.26 e	25.93 ± 2.36 a	4.96 ± 1.06 d	7.83 ± 0.86 c
	Amount	44.42 ± 3.64 b	96.67 ± 3.58 a	96.16 ± 4.62 a	48.39 ± 4.68 b	96.38 ± 4.88 a	96.06 ± 3.88 a
NH_4_^+^-N	0 h–8 h	13.19 ± 1.56 d	80.39 ± 4.58 c	91.36 ± 3.58 a	11.11 ± 1.26 e	83.97 ± 3.33 b	91.25 ± 3.62 a
	8 h–1 d	10.42 ± 1.52 b	8.53 ± 1.25 c	2.19 ± 0.62 e	11.81 ± 0.82 a	6.88 ± 1.58 d	2.05 ± 0.62 e
	1 d–2 d	20.83 ± 1.02 b	7.67 ± 0.88 c	1.96 ± 0.44 e	39.24 ± 3.58 a	5.74 ± 1.68 d	1.99 ± 0.40 e
	Amount	44.44 ± 3.25 c	96.59 ± 5.45 a	95.51 ± 4.22 a	62.15 ± 4.58 b	96.59 ± 5.88 a	95.28 ± 4.89 a
NO_3_^−^-N	0 h–8 h	−28.92 ± 4.50 d	28.61 ± 3.69 b	40.49 ± 4.58 a	−27.71 ± 3.58 d	32.95 ± 2.56 b	22.61 ± 2.66 c
	8 h–1 d	37.95 ± 3.68 c	60.54 ± 4.55 a	53.64 ± 3.69 b	10.64 ± 1.23 d	58.62 ± 4.68 a	55.94 ± 4.66 ab
	1 d–2 d	7.43 ± 1.52 b	1.02 ± 0.26 c	−1.66 ± 0.50 e	19.28 ± 1.60 a	0.00 ± 0.80 d	−1.28 ± 0.68 e
	Amount	16.47 ± 3.60 c	90.17 ± 5.20 a	92.46 ± 4.60 a	2.21 ± 0.88 d	91.57 ± 5.90 a	77.27 ± 5.55 b

^1^ The letters in the table indicate the different treatments of the same nitrogen form for a certain time after perfusion significantly (*p* < 0.05).

**Table 3 plants-13-00562-t003:** TN loss rate and absorption rate in different treatments.

Treatment	Nitrogen Application ^1^ (kg/hm^2^)	Loss Load of TN (kg/hm^2^)	Loss Rate of TN ^2^ (‰)	TN Content in Plants (kg/hm^2^)	TN Absorption Rate ^3^ (%)	Yield (kg/hm^2^)
FW	254.81 ± 0.00 b ^4^	2.70 ± 0.03 a	10.59 ± 0.39 a	167.65 ± 1.73 c	65.79 ± 0.68 b	8104.01 ± 248.12 b
FA	312.21 ± 0.00 a	2.70 ± 0.03 a	8.64 ± 0.38 c	231.19 ± 1.24 a	74.05 ± 0.40 a	9198.57 ± 335.02 a
FA-80	253.11 ± 0.00 b	2.38 ± 0.03 b	9.40 ± 0.22 b	184.22 ± 0.59 b	72.78 ± 0.24 a	7901.11 ± 168.73 bc
IW	254.81 ± 0.00 b	2.13 ± 0.03 c	8.36 ± 0.52 c	127.35 ± 1.83 d	49.98 ± 0.72 c	7630.41 ± 126.51 c
IA	307.02 ± 0.00 a	1.50 ± 0.02 e	4.87 ± 0.12 e	158.24 ± 1.80 c	51.54 ± 0.06 c	7531.36 ± 115.84 c
IA-80	250.86 ± 0.00 b	1.58 ± 0.02 d	6.29 ± 0.26 d	112.80 ± 1.47 e	44.97 ± 0.58 d	5742.89 ± 98.88 d

^1^ Nitrogen application includes fertilizer application and nitrogen provided by aquaculture wastewater; ^2^ TN loss rate means the ratio of loss load to nitrogen application; ^3^ Nitrogen absorption rate means the ratio of TN content in plants to nitrogen application; ^4^ Different letters indicate significant differences between treatments at the 0.05 level.

## Data Availability

All date included in this study are available upon request by contact with the corresponding author.

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
