# Peer review of "Nitrogen Loss and Migration in Rice Fields under Different Water and Fertilizer Modes"

_plants, 2024, doi:10.3390/plants13050562_

Round 1

Reviewer 1 Report

Comments and Suggestions for Authors

Authors,

The manuscript suggests a possibility of practical use of aquaculture wastewater (AW) with an observation of N dynamics. Based on the results, AW is available for rice production, whereas N dynamics in terms of degradation, migration and loss are not enough for readers to fully understand.

Please find the attached for reviewer's comments and suggestions.

Author Response

Thanks very much for taking your time to review this manuscript. We hope that the revised manuscript is closer to publication in the Plants. Once again, thank you very much for editors and reviewers’ warm work earnestly. I have responded separately to each of your comments, please see the attachment. The current occasion is the Chinese Lunar New Year, we wish the editors and reviewers a happy New Year.

Reviewer 2 Report

Comments and Suggestions for Authors

1. The abstract and conclusion are too long, and only the main results and conclusions need to be summarized.

2. I'm not sure why all the charts show only the mean value, but no standard deviation or standard error.

3. The discussion section should acknowledge the shortcomings and prospects of this study.

Author Response

Thanks very much for taking your time to review this manuscript. We hope that the revised manuscript is closer to publication in the Plants. Once again, thank you very much for editors and reviewers’ warm work earnestly. We have responded separately to each of your comments, please see the attachment. The current occasion is the Chinese Lunar New Year, we wish the editors and reviewers a happy New Year.

Reviewer 3 Report

Comments and Suggestions for Authors

RE: Nitrogen loss and migration in rice fields under different

This research study presents valuable insights into the potential of using aquaculture wastewater irrigation for sustainable rice production. The manuscript has the potential to be published following the comments and suggestion given below.

·       The opening statement of the abstract “Irrigating aquaculture wastewater in paddy fields reduces water and fertilizer loss” needs to be check how irrigation water can reduce the fertilizer losses, may be the optimum irrigation water can do.

·       No proper treatments and specific methodology are provided in the abstract section, please provide them

·       The concluding section, of the abstract “The findings can offer theoretical and technical direction for the purification of aquaculture effluent in paddy fields.” Is not clear, please re-structure the statement to make more concise conclusion and recommendation

·       The key words should be revised, and be related to the study

·       Clearly introduces the research topic and its importance and provide relevant background information on aquaculture wastewater and rice production.

·       Consider mentioning specific examples of environmental and economic concerns associated with traditional wastewater disposal methods.

·       Procedure for measuring various parameters is not given, please provide the brief procedure how you determined the various parameters, please also provide a valid reference for The calculation of nitrogen reduction rate(θ1)

·       Briefly touch upon the potential benefits of using AW irrigation for sustainable agriculture which may improve the importance of the study.

·       The results section needs to be improved by providing the key results in a well-organized and easy-to-understand manner, and to compare the different treatments, it is not necessary to write each and every things been presented in the tables/figures

·       Highlights key observations regarding nitrogen degradation, migration, loss, and absorption.

·       Consider including specific numerical values preferably in % increase or decrease in the text alongside figures and tables for enhanced readability.

·       Briefly explain the potential mechanisms behind the observed differences in nitrogen degradation rates under different irrigation and fertilization treatments.

·       Discuss the limitations of the chosen methods and acknowledge potential sources of error in the data.

·       Discusses the findings in relation to existing literature and provides context with practical recommendations for farmers based on the research findings.

·       Consider quantifying the economic and environmental benefits of AW irrigation compared to traditional methods.

·       Explore the potential negative impacts of long-term AW irrigation on soil microbial communities and trace element contamination.

·       Develop specific recommendations for farmers based on different soil types, climatic conditions, and rice varieties.

·       Briefly touch upon the potential mechanisms behind the differences in nitrogen migration under different irrigation modes.

·       Concisely summarizes only the key findings of the study with clear implications for both research and practice. That should contain a take home message

·       Fig 1, the x-axis should be formatted give full range of crop duration and precipitation should be provided in total per month, Fig 3 is much confusing and not readable, Fig 4 is not clear, the quality of all figures needs to be improved

·       Table 1 is not clear, please shift to pictorial illustration or text, Table 2, the various abbreviations used in 2nd line of the table should be explained below the table, the decimal points should be rounded to 1 or 0 in table 4

Comments on the Quality of English Language

Moderate editing is needed 

Author Response

(The authors gave the same response as above.)

Round 2

Reviewer 3 Report

Comments and Suggestions for Authors

The authors have incoporated most of my comments, and thus I am accepting the manuscirpt in its current form 

Comments on the Quality of English Language

The English need minor editing for fluency